# Identification of Laminar Composition in Cerebral Cortex Using Low-Resolution Magnetic Resonance Images and Trust Region Optimization Algorithm

**DOI:** 10.3390/diagnostics12010024

**Published:** 2021-12-23

**Authors:** Jakub Jamárik, Lubomír Vojtíšek, Vendula Churová, Tomáš Kašpárek, Daniel Schwarz

**Affiliations:** 1Department of Psychiatry, Faculty of Medicine, Masaryk University, 625 00 Brno, Czech Republic; 461084@muni.cz (J.J.); tkasparek@med.muni.cz (T.K.); 2Neuroscience Centre, Central European Institute of Technology, Masaryk University, 625 00 Brno, Czech Republic; vojtisek@ceitec.muni.cz; 3Department of Simulation Medicine, Institute of Biostatistics and Analyses, Faculty of Medicine, Masaryk University, 625 00 Brno, Czech Republic; churova@biostatistika.cz

**Keywords:** cortical layers, mathematical modeling, MR imaging, optimization algorithm, brain imaging

## Abstract

Pathological changes in the cortical lamina can cause several mental disorders. Visualization of these changes in vivo would enhance their diagnostics. Recently a framework for visualizing cortical structures by magnetic resonance imaging (MRI) has emerged. This is based on mathematical modeling of multi-component *T*_1_ relaxation at the sub-voxel level. This work proposes a new approach for their estimation. The approach is validated using simulated data. Sixteen MRI experiments were carried out on healthy volunteers. A modified echo-planar imaging (EPI) sequence was used to acquire 105 individual volumes. Data simulating the images were created, serving as the ground truth. The model was fitted to the data using a modified Trust Region algorithm. In single voxel experiments, the estimation accuracy of the *T*_1_ relaxation times depended on the number of optimization starting points and the level of noise. A single starting point resulted in a mean percentage error (MPE) of 6.1%, while 100 starting points resulted in a perfect fit. The MPE was <5% for the signal-to-noise ratio (SNR) ≥ 38 dB. Concerning multiple voxel experiments, the MPE was <5% for all components. Estimation of *T*_1_ relaxation times can be achieved using the modified algorithm with MPE < 5%.

## 1. Introduction

The highest functions of the brain are enabled by the complex functional architecture of the cerebral cortex. Therefore, it is no surprise that pathological malformations within the cortex can lead to various disorders. These pathological changes occur at resolutions significantly lower than the ones available to the current neuroimaging hardware, causing an obstacle to their direct diagnosis in vivo.

The structure of the cerebral cortex was first extensively examined ex vivo [1], resulting in a description of its laminar architecture, commonly separated into six cortical layers [2]. However, the sub-millimeter thickness of the cortical lamina prevents equivalent visualization in vivo. First attempts to circumvent this limitation focused on anatomically distinct formations within the cortex. Such a structure is the stria of Gennari—a strongly myelinated stripe located within layer IV of the primary visual cortex. Positioning the imaging slices perpendicular to the region of interest allowed Clark et al. [3] to capture the stria in black contrast, using field strengths of 1.5 T. Additional work followed targeting this cortical landmark. The researchers employed higher field strengths of 3 T and the acquisition of multiple images of the same subject. These were used for averaging, which was necessary to achieve an appropriate signal-to-noise ratio (SNR) needed for visualization [4,5,6,7]. Further development was needed to decrease the total imaging duration caused by the image averaging and improve the image contrast. These goals were accomplished using magnetization-prepared rapid gradient-echo (MPRAGE) and gradient-echo (GE) sequences at field strengths of 7 T [8,9,10]. While the imaging of cortical layers began with a focus on primary visual cortex V1, additional areas of cortical lamina followed. Researchers focused on the motion-sensitive area V5 [11] and the auditory cortex [12], acquiring T_1_-weighted images at 3 T. Due to the sub-millimeter resolutions possible at 7 T, the focus of the research community shifted to image acquisition at the highest field strengths. Multiple Brodmann areas of the cortex were measured using a magnetization-prepared fluid-attenuated inversion recovery sequence. The result was several intensity profiles, which exhibited a multiple-layer appearance similar to the patterns of the cortical lamination [13]. Different contrasts resulting from a modified magnetization-prepared rapid acquisition GE sequence were combined to create intercortical maps related to myelin content. Subsequent clustering yielded a delineation of the auditory area [14]. Laminar profiles resembling the lines of Baillarger were also revealed in the images resulting from a modified T_1_-weighted MPRAGE sequence [15]. Magnetization-prepared sequences of two rapid acquisition gradient-echoes (MP2RAGE) were used to acquire high-resolution T_1_-weighted images. The cortical gray matter was segmented out of the volume and then segmented further, revealing four cortical layers [16]. A conceptually different approach was used to visualize cortical layers without the necessity of sub-millimeter image resolution. A fast spin-echo (SE) sequence with several different IR times at 3 T captured several images with corresponding contrasts. The dataset was then fitted to an exponential decay function to estimate the *T*_1_ relaxation times individually for each voxel. The estimated values served as the basis for the classifications of individual voxel into five or six groups, corresponding to the cortical layers [17]. Using a similar imaging protocol, a series of low-resolution echo-planar images (3 mm) were acquired with the contrast based on a set of varying IR times. A modified fitting procedure allowed for the estimation of multiple *T*_1_ relaxation times related to individual voxel components, thus capturing several layers within a single voxel [18]. The above-mentioned imaging procedure was also made to better reflect the natural curvature of the cerebral cortex. This was accomplished via sub-sampling of individual voxels and their mapping onto a grid of virtual spheres, spanning the cortical gray matter [19]. The works presented so far show two emerging pathways in the imaging of whole-brain cortical lamination. The first approach is focused on acquisitions of high-resolution images at higher field strengths (7 T respectively) [14,15,16]. Although utilized in a variety of research endeavors, this approach is not without limitations, the most notable being the partial volume effect (PVE). This is the occurrence of multiple tissue types within a single voxel, which manifest in the obtained voxel intensity [20]. In the context of cortical laminations, this effect persists even at 7 T [18]. An alternative approach to imaging the cortical layers is based on the acquisition of a multitude of images—surprisingly—with lower resolutions at lower field strengths. The low-resolution images are subjected to a complex modeling and visualization pipeline resulting in high-detail maps of cortical lamination. This approach is limited due to the need for estimation of *T*_1_ relaxation times, the process of which is a tradeoff between computational complexity, time constraints, and estimation accuracy [18,19].

In this paper, we use the existing low-resolution approach to imaging cortical layers. We endeavor to increase the accuracy of the mathematical modeling, which forms an integral part of the overall method. More specifically, we investigate whether the Trust Region algorithm is able to estimate the *T*_1_ values of several components within a single voxel image using the pulse sequence proposed in [18]. A dataset with known values of *T*_1_ times is generated to assess the validity of the method. This is achieved via simulations of MRI images and individual voxels. Simulations are carried out using signal equations and an established simulator MRiLab, with a custom sequence and an imaging phantom.

The remainder of the paper is organized as follows. In the Materials and Methods, we first describe the estimation of *T*_1_ relaxation times as an optimization problem and describe the chosen algorithm. Later, we focus on the description of the experimental and simulated data. In the Results section, we present the outcomes of the optimization algorithm for various levels of noise and types of simulated data. The Discussion compares the results with results of similar research endeavors in estimating *T*_1_ relaxation and concludes the paper.

## 2. Materials and Methods

### 2.1. Fitting Problem

Assuming that the time of repetition $\gg T_{1}$ and the first RF pulse is equal to 180°, the equation for GE inversion recovery and SE inversion recovery sequences, which models a single voxel signal, can be formulated as:(1)$$S\left(TI_{i}\right)=c\left(1-2e^{-\frac{TI_{i}}{T_{1}}}\right),$$
where $S\left(TI_{i}\right)$ is complex-valued and represents single-voxel image information after the Fourier transform, $TI_{i}$ is the time of inversion for the *i*-th inversion recovery time, and *c* is complex-valued. The voxel intensity is dominantly weighted by the relaxation time *T*_1_ but is also influenced by other relaxation mechanisms. A more generalized form of Equation (1) can be used to estimate the *T*_1_ relaxation times of a single signal source. This is usually a single voxel of an MR image, commonly used for *T*_1_ mapping, as evidenced by the state-of-the-art method [21].

The imaging protocol proposed in [18] produces only magnitude images after the inverse discreet Fourier transform. Hence, we have to limit the model to the magnitude data. Assuming only signal magnitude is available, Equation (1) takes the form of:(2)$$\left|S\left(TI_{i}\right)\right|=M_{0}\left|\left(1-2e^{-\frac{TI_{i}}{T_{1}}}\right)\right|,$$
where $\left|S\left(TI_{i}\right)\right|$ is voxel intensity, $\left|c\right|=M_{0}$, and the parameter $M_{0}$ denoting magnetization corresponding to the center of the k-space for the given voxel at TI=0 ms. This model can be generalized to include multiple components per voxel [18]. In that case, Equation (2) takes the form of:(3)$$M\left(TI_{i}\right)=\overset{n}{\underset{j=1}{\sum }}M_{0j}\left|\left(1-2e^{-\frac{TI_{i}}{T_{1j}}}\right)\right|,$$
where the magnetization of the voxel for the *i*-th inversion recovery time $M\left(TI_{i}\right)$ is equal to the sum of individual magnetizations of the assumed components, $M_{0j}$ is the magnetization at TI=0 ms for the *j*-th component and $T_{1j}$ is the *T*_1_ relaxation time for the *j*-th component, and *n* denotes the number of components per voxel. While the parameter *T*_1_ uniquely identifies the cortical component, the parameter *M*_0_ is proportional to the relative representation of the cortical component within the voxel.

The modeling problem encountered here could be classified under the domain of the multiexponential analysis [22]. It is often encountered in material science as a part of nuclear magnetic resonance (NMR) relaxometry [23]. The NMR signal is decomposed based on the properties (relaxation times) of the individual structural elements within the measured sample, revealing their relative composition [24]. This is achieved via the inverse Laplace transform [25], although this term is also used to describe mathematically distant methods [26]. The result is a distribution of relaxation times for each voxel. In our work, we follow a different path of multiexponential analysis but with a similar goal of identifying the underlining components that modulate the obtained signal.

To estimate the coefficients, $M_{0j}$ and $T_{1j}$, model (3) is fitted to measured data (see Figure 1) using the non-linear least-squares method.

The data in this context represent magnitudes of a single voxel at the same position within all of the images acquired with different inversion recovery times. The objective function takes the following form:(4)$$F\left(M_{01},M_{02},\ldots ,M_{07},T_{11},T_{12},\ldots ,T_{17}\right)=\overset{n}{\underset{i=1}{\sum }}{\left(M\left(TI_{i}\right)-\overset{7}{\underset{j=1}{\sum }}M_{0j}\left|\left(1-2e^{-\frac{TI_{i}}{T_{1j}}}\right)\right|\right)}^{2},$$
where $TI_{i}$ represents the duration of inversion recovery in ms for i=1,2,…,n for *n* images. The final step is the minimization of the objective function using a suitable optimization algorithm. For this purpose, we chose the Trust Region algorithm with a modification in its initialization. The algorithm, including the modification, is described in the following sections.

### 2.2. Experimental MRI Data

The measured datasets resulted from 16 experiments using a 3T Siemens Magneton Prisma MRI scanner (Siemens Healthcare, Erlangen, Germany). Each experiment consisted of two different imaging protocols:Low-resolution modified echo-planar sequence with the following parameters: TR/TE = 1200/39 ms, 105 inversion times from the interval of 50–3000 ms, with the resolution of 3 × 3 × 3 mm^3^. The size of the image obtained from this sequence was 64 × 64 × 42 voxels.High-resolution MPRAGE sequence with the following parameters: TR/TE = 2150/2.5 ms, TI = 1100 ms, with the resolution of 1 × 1 × 1 mm^3^. The size of the image obtained from this sequence was 160 × 256 × 256 voxels.

### 2.3. Simulated Data

Additional imaging data were also generated to provide ground truth information to enable assessment of the ability of our optimization algorithm to arrive at the best solution. Ten different imaging datasets were simulated in total (using Matlab (v2019a, Mathworks, Natick, MA, USA)), each with an increasing level of complexity.

The first nine simulated datasets consisted of a series of intensity magnitudes of a single voxel, proportional to the magnetization $M\left(TI_{i}\right)$. These values were computed using Equation (3). Parameters $M_{0j}$ and $T_{1j}$ were selected for all cortical components, and the values of $M\left(TI_{i}\right)$ for the inversion times from 50–3000 ms were computed, based on this selection.

Values for the parameter *T*_1_ were chosen based on the whole-brain estimates, using the method presented in [21]. A single *T*_1_ value was obtained for each voxel of the whole image. The histogram of the relaxation times is presented in Figure 2.

As shown in Figure 2, the data show two distinct peaks corresponding to the *T*_1_ times of 700 ms and 1000 ms. These represent the *T*_1_ relaxation times of white (700 ms) and gray (1000 ms) matter at 3 T, similar to the values found in the literature [27,28].

While the cerebral cortex is commonly delineated into six layers, our model (3) assumes seven individual components. Given the resolution of our data, it is reasonable to assume the presence of additional bordering structures (cerebrospinal fluid (CSF) and white matter (WM)) within the cortical voxels. Therefore, up to seven unique components per voxel were assumed, each representing a unique formation within the cerebral cortex (CSF, WM, the sixth cortical layer, and the five remaining cortical layers).

For the first simulated dataset, the values of parameter *M*_0_ were randomly generated while the values of parameters *T*_1_ (700, 800, 1100, 1200, 1500, 1700, and 2000 ms) remained constant. Parameters *M*_0_ were generated from a uniform distribution to ensure a minimum representation of 5% of each component in the voxel. To make the simulated data more closely resemble the outcome of the MRI experiments, a noise component with the Gaussian distribution with the expected value equal to zero and increasing variance ($\sigma^{2}=\left\{0,\;0.1,\;1,\;5,\;10,\;25,\;50,\;100\right\}$) was added. This process resulted in eight datasets with varying levels of noise. An illustration describing the generation of the simulated data is presented in Figure 3.

The final simulated dataset was created using the MRiLab [29] simulator (v1.3, Fang Liu, Madison, WI, USA). The software allows for the Bloch equation-based simulation of the MRI process. Modeling of the tissue microstructure at the sub-voxel level is achieved by using the generalized tissue model with multiple proton exchange pools.

A virtual object with the predefined properties and an MRI sequence (multi-shot EPI with *TR* = 10 s, *TE =* 30 ms), similar to the sequence used for the measurement of the real-life data, was created. The virtual object was 100 × 100 × 32 voxels and consisted of two components (*T*_11_ = 700 ms, *T*_21_ = 80 ms, *ρ*_1_
*=* 0.4 *T*_12_
*=* 800 ms, *T*_22_ = 90 ms, *ρ*_2_
*=* 0.6). Given the limitations of the simulator, only two components per voxel could be simulated. The size of the virtual object was chosen based on the properties of the virtual objects supplied with the simulator. Using the defined virtual object and imaging protocol, a series of MRI experiments with different inversion recovery times from 50 ms to 960 ms were simulated. The result was a dataset consisting of 70 images (64 × 64 pixels).

### 2.4. Modified Trust Region Algorithm

Trust Region algorithms represent a set of optimization algorithms based on the approximation of the objective function within a selected region of the optimization search space [30]. The approximating function should be able to reasonably represent the objective function within the selected region and be easier to optimize. The algorithm itself iteratively finds a local solution to a problem:(5)$$min_{x\in R^{n}}\;\;f\left(x\right),$$
where f(x) is a twice-differentiable vector-valued objective function. For the *k*-th iteration of a general Trust Region algorithm, the first step is the approximation of the objective function. A model $m_{k}\left(x\right)$ is established to approximate the objective function within a neighborhood of $x_{k}$. A commonly used approximation is the Taylor series expansion of f(x) around the point $x_{k}$:(6)$$m_{k}\left(x_{k}+s\right)=f\left(x_{k}\right)+g_{k}{}^{T}s+\frac{1}{2}s^{T}H_{k}s,$$
where $g_{k}$ denotes the gradient of the objective function and $H_{k}$ is the Hessian. The trust-region $B_{k}$ for the *k*-th iteration can be then defined as:(7)$$B_{k}=\left\{x\in R^{n}|\|x-x_{k}\|{}_{k}\leq \Delta_{k}\right\},$$
where $\Delta_{k}$ is the trust-region radius and $\|\cdot \|{}_{k}$ is an iteration-dependent norm, often the $l_{2}$ norm. Using the model $m_{k}\left(x\right)$, a trial step $s_{k}$ is estimated to satisfy the condition $\|s_{k}\|{}_{k}\leq \Delta_{k}$. This is known as the trust-region sub-problem, and its successful solution results in the computation of the trial point $x_{k}+s_{k}$. The ability of the model $m_{k}\left(x\right)$ to predict the change in the objective function is then assessed by comparing the relative change of f(x) and $m_{k}\left(x\right)$. If the result is within a predefined threshold, the trial point is accepted and the trust-region remains the same or increases. Otherwise, the trust-region is decreased and the trial point is computed again.

Apart from the choice of the approximating model, a way of solving Equation (6) must also be specified. In the case of large bound-constrained problems, an effective approach for solving the model equation is the restriction of the trust-region into a two-dimensional subspace [31,32]. The two-dimensional subspace is assumed to be spanned by two vectors. These vectors correspond to the direction of the negative gradient and either the Newton direction (vector v solving the equation H.v=−g) or to a direction of the negative curvature (vector v solving the equation $v^{T}H.v<0$).

For our work, the algorithm mentioned above is modified by adding multiple starting points. The optimization algorithm is repeatedly initialized from several randomly chosen starting points. This modification prevents incorrect termination of the optimization procedure in local nonoptimality traps.

## 3. Results

The method’s performance described in the previous section was evaluated on the simulated imaging datasets (see Figure 3). Initialization of the optimization procedure was assessed using the first dataset. The whole process of simulating and fitting data was repeated 10 times. The results are presented in Table 1.

The mean value of mean squared error across 10 repetitions was 0.193 (4.43*e*^−27^ for 100 starting points) with a standard deviation of 0.288 (1.46*e*^−27^ for 100 starting points). This contrasts with the values from Table 1, which show an error for individual coefficients up to several hundred percent. Table 1 also shows the difference between the numbers of starting points used to initiate the optimization and the relative error of coefficients. In the case of a single starting point, the maximum relative error for *M*_0_ was 604% (ground truth = 61.8, estimated = 435.1), while the optimization with 100 starting points resulted in 0% relative error.

In the next step, the model was fitted to the datasets with an added noise component. No prior knowledge of parameters was assumed, only the minimum representation of 5% for each component per voxel. The values of coefficients to be estimated were bounded using the maximum values in data for *M*_0_ and the estimates of *T*_1_ relaxation times in the literature [28] (fat $T_{1}=250\;\mathrm{ms}$, CSF $T_{1}=4000\;\mathrm{ms}$). The results are presented in Table 2.

Table 2 demonstrates that the value of relative errors in estimated coefficients increases with the decrease in SNR. This is even more visible in the case of the coefficients *M*_0_. The mean relative error for the coefficients *M*_0_ at the SNR level of 45 dB is 28%, while the maximum relative error of a single coefficient is 109%. The relative error of the coefficients *T*_1_ is lower than the relative error of the coefficients *M*_0_. At the SNR level of 31 dB, the mean relative error of coefficients *T*_1_ is 11%, while the maximum relative error of a single coefficient is 26%.

The final fit of the model was conducted using the multiple voxel-simulated images generated by the MRiLab simulator. The data in this context are represented by the magnitudes of each pixel across the 70 images with different *TI* times. The coefficients to be estimated were bounded using the maximal value of magnitude in data for parameter *M*_0_ and estimates of *T*_1_ relaxation times in the literature for parameter *T*_1_. The histogram of the estimated coefficients *T*_1_ for all voxels representing the virtual object is presented in Figure 4.

Figure 4 illustrates that the majority of the estimates were between 700 and 800 ms. Several notable peaks were present, mostly around values of 750 ms and 770 ms. A noticeable gap is present at 760 ms, indicating two separate distributions for the parameter estimates (for 700 ms and 800 ms, respectively). The distributions are skewed towards the estimation average. Table 3 shows the relative error of the estimated coefficients.

The mean error of the first component (4.86%) is higher than the mean error of the second component (2.98%) by 1.88%. The minimum relative error of the second component (0.12%) is more than two times larger than the first component’s error (0.05%). It can be concluded that the estimation of coefficient *T*_1_ for the first component is less precise than the estimation for the second component.

## 4. Discussion

Current MR imaging of cortical layers mostly focuses on increasing the spatial resolution of the images [14,15,16]. Alternatives arise when attempting to capture the patterns of cortical lamination in the domain of spin-lattice relaxation [18,19]. This approach forgoes the requirements for high spatial resolution, demanding a higher number of images instead. The high volume of images, and, therefore, data points, is needed for subsequent signal processing and analysis. The crucial point of the imaging method lies in mathematical modeling—estimating the underlying composition of every voxel. The visualization approaches following the processing [18] rely on the precise estimation of the *T*_1_ values of each cortical component/layer.

The low-resolution modeling approach commonly employed stems from fMRI *T*_1_ mapping. It relies on the Levenberg–Marquardt algorithm to solve the underlying optimization problem and estimate the relaxation times [21]. The method has a simplified form used when only magnitude data are available, a procedure named polarity restoration. This process stands for the inversion of select data, circumventing the restrictions posed by the magnitude-only data on the objective function. The function itself has only two parameters per voxel, which leads to a 2D search for the optimal solution. This is further eased by restricting the possible values of *T*_1_ times to whole numbers between 1 and 5000 ms. As a direct result, a grid search coupled with a 1D search of the L–M algorithm can be used, increasing the precision of the resulting estimate.

However, using this methodology for multiple sub-voxel components is not a straightforward extrapolation of the original modeling framework. Assuming up to 14 parameters per voxel makes a derivation of a model similar to the one previously used [21] much more complex. The lack of such a model would also require a different approach for the polarity estimation. The simplification of bounding the *T*_1_ times within a predefined interval would also cause a substantial increase in the estimation time. The size of the search grid would expand to 5000^7^ points, and a 7D search would still need to be performed from all the points.

Our approach to low-resolution modeling differs by using only the magnitude data with no polarity restoration. We explore whether the Trust Region algorithm can solve the proposed optimization problem with fewer computational demands. The optimization results on the first dataset show that the use of the unmodified Trust Region algorithm proved insufficient for the model in question. The estimation of the non-linear parameters (coefficients of relaxation time *T*_1_) increased the complexity of the model and likely introduced nonoptimality traps. To circumvent this limitation, we modified the algorithm in question. The search space for each variable was bounded, and the optimization procedure was initiated from multiple starting points, which resulted in the ideal fit of the model.

While the mean squared error (MSE) value was comparatively low, the estimated values of the model coefficients differed more substantially. Therefore, it can be concluded that the sole use of MSE is insufficient for exhaustive measures of the model performance. It is more suitable to evaluate the error rates of the individual coefficient estimates.

The inclusion of noise with the Gaussian distribution allowed further assessment of the model robustness. The noise distribution was chosen based on the distributions used in MRI simulation experiments [29]. It should be noted that the inclusion of the Gaussian noise is usually incorporated after the formation of the image k-space during the simulation procedure. The final image usually contains noise with Rician or Rayleigh distribution [33,34]. As stated previously, the most suitable method for assessing the model performance is to evaluate the individual coefficient estimates. As expected, the error rate for individual coefficients increases with the noise level. This increase seems to be higher for the coefficients *M*_0_ than the coefficients *T*_1_. It could be concluded that the method used is more efficient in the distinction of individual exponential curves (parameter *T*_1_) than in estimating their relative representation (proportional to *M*_0_) within the voxels.

With the final simulated dataset, the performance of the chosen modeling approach on data more closely resembling real images could be examined. The model of choice had to be reduced to incorporate two components due to the limitations of the utilized existing MRI data simulator. The limits of the simulator also affected the choice of *TI* time points, the number of which had to be reduced. The modeling outcomes on the simulated datasets showed a successful application of the method with the mean error of *T*_1_ coefficients estimates under 5%. Estimates of the *M*_0_ coefficients could not be fully examined as there is no precise relationship to the proton density ρ used to create the virtual object.

The outcomes acquired from the last dataset more closely illustrate the various tradeoffs accompanying the proposed method. To increase the precision of the estimation, a sufficient number of images acquired with different *TI* times is required. However, it also causes longer scan time, which increases the risk of subject movement. This can be possibly remedied by image registration before the modeling itself. Another limitation of the proposed approach is the requirement of high SNR needed for a sufficiently precise estimation of the *T*_1_ values.

Direct imaging of all cortical layers is not yet feasible even with ultra-high field MRI (7 T and higher), although some of the layers can be distinguished [16,18]. This paper follows the ideas presented in [18], i.e., the delineation of the cortical layers is based on the ability to quantify the intravoxel concentration of compounds with different *T*_1_ relaxation properties. With this approach, it is possible to acquire a proportional representation of the cortical layers in the domain of *T*_1_ relaxation, not the spatial domain. However, at this stage, the method is still highly experimental. Its introduction into preclinical studies will not be feasible without first assessing the ability of the method to reliably estimate the proportional representation of the layers. Further steps prior to clinical studies of a diagnostic application may include (i) evaluation of correlations between the identified MR imaging parameters of the cortical layers and selected variables resulting from histology of animal models, (ii) comparison with other, established methods such as voxel-based morphometry, deformation-based morphometry, or diffusion kurtosis imaging. Given the possible clinical application in diagnostics, it is difficult to imagine this modality being used in isolation. Rather, it will be part of a multimodal approach to cortical pathology imaging, as it has the potential to provide complementary information on the internal cortex arrangement in pathologies that have a complex morphological correlate involving changes in elemental complexity, such as neuronal atrophy, dendritic tree reductions with increased density of neuronal bodies, migration of activated microglia, etc. We currently employ an onsite modified pulse sequence to ensure full control of the sequence parameters, over all TIs, to ensure reliable intra-voxel multiple *T*_1_ fitting. We are aware of several limitations related to the parameters of a real non-ideal pulse sequence, such as nonzero TE and finite TR. At this stage, we aim for replicable imaging with sufficient sensitivity to identify laminar cortical layer composition, not a quantitative measurement.

We consider the aforementioned estimations, i.e., the results of mathematical modeling, to be a critical part of this new low-resolution approach to imaging the laminar structure of the cerebral cortex. We believe that using the modified Trust Region algorithm dramatically improves the overall method. The decreased computational time and less severe requirements on the objective function could reduce entry barriers for wider acceptance of the low-resolution imaging framework. This could aid its adoption by potential researchers and further development for use in clinical diagnostics.

## Figures and Tables

**Figure 1 diagnostics-12-00024-f001:**
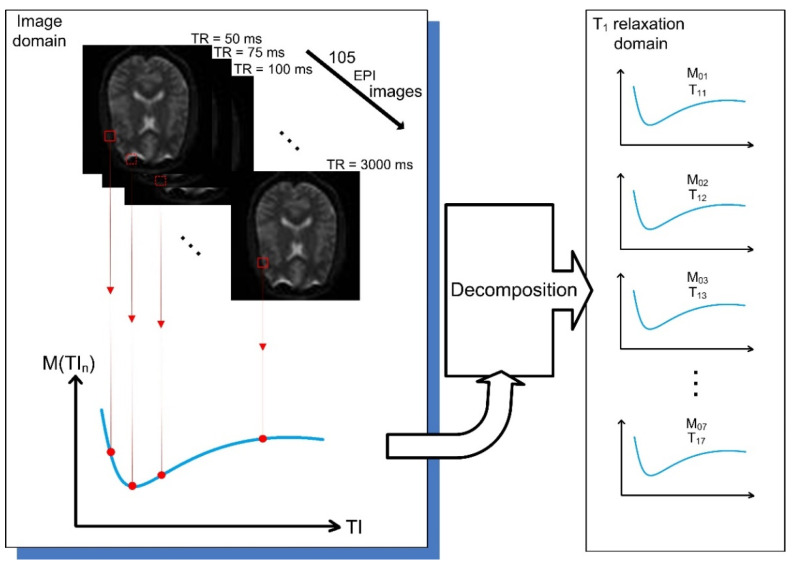
Illustration of the data acquisition approach proposed by Lifshits et al. [18]. The experimental data consist of a series of EPI images with different times of inversion. A one-dimensional signal is constructed for every image voxel, dependent on the *TI* time. The signal is then decomposed into several curves, each representing a voxel component with specific values of *M*_0_ and *T*_1_. In this way, a cortical composition of a single voxel series can be decomposed into multiple signals in the *T*_1_ relaxation domain.

**Figure 2 diagnostics-12-00024-f002:**
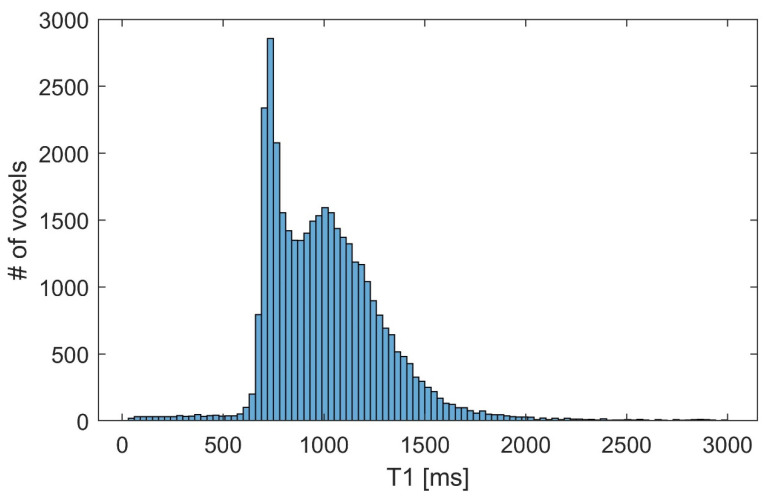
Histogram of estimated *T*_1_ values, one per voxel, whole-brain image.

**Figure 3 diagnostics-12-00024-f003:**
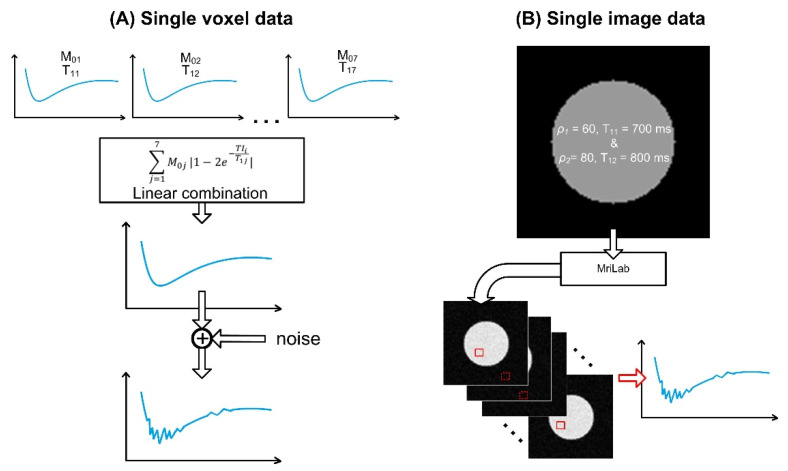
Generating simulated data. (**A**) Data simulating the signal of a single-voxel series. Individual signals with chosen parameters *M*_0_ and *T*_1_ are linearly combined, and a noise component of varying power is added. The result is a signal resembling a magnitude series of a single voxel from the experimental data. (**B**) Simulation of a 2D image. A numerical phantom with two components per voxel is constructed. It is then subject to the experimental EPI sequence, resulting in a series of images.

**Figure 4 diagnostics-12-00024-f004:**
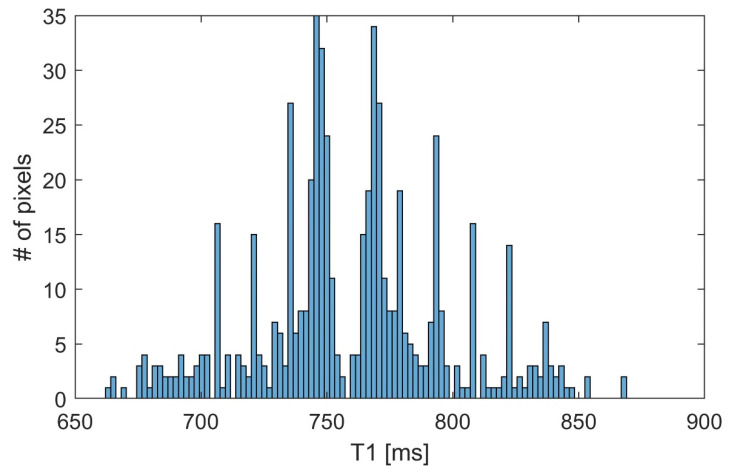
Histogram of estimated coefficients *T*_1_.

**Table 1 diagnostics-12-00024-t001:** Relative error of estimated coefficients—comparison of multiple starting points.

	*N_starting points_* = 1			*N_starting points_* = 100		
	Min. Error [%]	Mean Error [%]	Max. Error [%]	Min. Error [%]	Mean Error [%]	Max. Error [%]
*M* _0_	0.00	44.60	604.00	0.00	0.00	0.00
*T* _1_	0.00	6.11	36.00	0.00	0.00	0.00

Min.—minimum, Max.—maximum.

**Table 2 diagnostics-12-00024-t002:** Relative error of estimated coefficients—data with added noise.

Noise Variance	SNR [dB]	Min. *M*_0_ Error [%]	Mean *M*_0_ Error [%]	Max. *M*_0_ Error [%]	Min. *T*_1_ Error [%]	Mean *T*_1_ Error [%]	Max. *T*_1_ Error [%]
0	Inf	0	0	0	0	0	0
0.1	61	0	2	4	0	0	0
1	51	2	5	14	0	0	1
5	45	2	28	109	0	2	4
10	41	3	18	48	0	1	3
25	38	3	28	86	0	2	5
50	34	7	80	268	1	10	25
100	31	18	60	131	2	11	26

**Table 3 diagnostics-12-00024-t003:** Relative error of estimated relaxation times.

*T* _1_	Min. Error [%]	Mean Error [%]	Max. Error [%]
700	0.05	4.86	8.65
800	0.12	2.98	8.59

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
