# Peer review of "Identification of Laminar Composition in Cerebral Cortex Using Low-Resolution Magnetic Resonance Images and Trust Region Optimization Algorithm"

_diagnostics, 2021, doi:10.3390/diagnostics12010024_

Round 1

Reviewer 1 Report

This is a very clear and precise paper on an important topic in clinical neuroradiology. The authors try to present a method to improve the NMR-imaging of the cortical layer, with the final aim to resemble the histological picture as closely as possible.

Therefore they describe the hitherto established methods and suggest a new algorithm that is called modified Trust Region algorithm which is tested in several MRI-simulations. The results of the simulations are clearly presented and adequately discussed. The authors draw the conclusion that the trust-region algorithm could greatly improve the low-resolution-based imaging of the cortical layers.

Author Response

This is a very clear and precise paper on an important topic in clinical neuroradiology. The authors try to present a method to improve the NMR-imaging of the cortical layer, with the final aim to resemble the histological picture as closely as possible. Therefore they describe the hitherto established methods and suggest a new algorithm that is called modified Trust Region algorithm which is tested in several MRI-simulations. The results of the simulations are clearly presented and adequately discussed. The authors draw the conclusion that the trust-region algorithm could greatly improve the low-resolution-based imaging of the cortical layers.

Response: Dear reviewer, thank you for reading the first version of the manuscript and providing your positive and encouraging feedback. We believe that the manuscript is now even better after the changes made to take into account the comments of other reviewers.

Reviewer 2 Report

The authors reported a new approach to estimate the cortical laminal structure based on simulated data acquired from sixteen MRI experiments of healthy volunteers.

I had the impression that this study was a simulation for simulation's sake.

(1) The results would not be able to reveal the layered structure of the cerebral cortex.

(2) Further research is needed because this method is not likely to solve their objective.

(3) At least, they need to clarify the weaknesses of this method and how they can achieve their goal in the future.

(4) It is necessary to clarify how to make the method clinically usable and what is lacking technically.

Major issue:

# Figure 1

This figure shows the overview of the research, so it is necessary to explain the content of the study. It is difficult to grasp the whole picture of the study design from this figure and legend.

# Figure 2

1) Why is there a valley in the histogram at around 760ms? Since there is a peak of M0 at about 25ms, the signal in between is probably due to noise. Would you please clarify the meaning of this graph?

# Several keywords are not explained.

1) There is an explanation in the established simulator MRiLab: p5L167-172, but I think a brief description is needed.

2) p4L147, "equation", refer to citation #3, was not explained.

Minor issues:

#1 For equipment and software, you need to include the manufacturer’s information.1)MRI scanner: e.g. (Siemens Healthcare, Germany)

2) MRiLab (Manufacturer, city, Nation)

p3L134, Delete repeated "1".

p4L138, Delete repeated "2".

Author Response

Dear reviewer, thank you for your valuable comments and insights which made us improve our manuscript significantly. Some of your comments correlated with those of Reviewer_3, and the most important ones led us to change the name of the manuscript to Identification of Laminar Composition in Cerebral Cortex Using Low-Resolution Magnetic Resonance Images and Trust Region Optimization Algorithm. We believe that we have responded to all your comments.

Point 0: The authors reported a new approach to estimate the cortical laminal structure based on simulated data acquired from sixteen MRI experiments of healthy volunteers. I had the impression that this study was a simulation for simulation's sake.

Response 0: After reading this comment and other reviews, we realized that we had insufficiently explained the reason for including the simulated data. In an effort to replicate low-resolution imaging experiments in [18], we encountered a number of problems with experimentally measured data and wanted to verify the ability to find the optimal model solution on data free from noise and other artifacts. Driven by the effort to reduce an unbearably large number of degrees of freedom, we experimented with simulated data and then designed a modified Trust-Region optimization algorithm, which gives excellent results. We also comment on this in more detail in other reactions.

[18] Lifshits, S.; Tomer, O.; Shamir, I.; Barazany, D.; Tsarfaty, G.; Rosset, S.; Assaf, Y. Resolution Considerations in Imaging of the Cortical Layers. NeuroImage 2018, 164, 112–120, doi:10.1016/j.neuroimage.2017.02.086.

Point 1: The results would not be able to reveal the layered structure of the cerebral cortex.

Response 1: In this paper, we use the well-established concept of low-resolution imaging of cortical layers introduced in (Liftshits et al., 2018) and in several other published works. It should be noted that this approach allows obtaining proportional representation of cortical layers in the domain of spin-lattice relaxation, not the spatial domain. This is done assuming a direct relationship between the T1 relaxation of the cortical components and their layered structure. This is indeed achieved through their myeloarchitectonic properties – changes in the degree of myelination across cortical layers. However, this is not uniform across the whole brain – another challenging aspect for a low-resolution approach. Our work does not focus on the development of an MRI sequence capable of capturing this phenomenon; this work has already been presented in the literature [18]. We aim to increase the accuracy of the mathematical modeling that forms an integral part of the overall method. After the revised Introduction (p2r91-98) and Discussion (p11r357-377), our goals should now be clear.

[18] Lifshits, S.; Tomer, O.; Shamir, I.; Barazany, D.; Tsarfaty, G.; Rosset, S.; Assaf, Y. Resolution Considerations in Imaging of the Cortical Layers. NeuroImage 2018, 164, 112–120, doi:10.1016/j.neuroimage.2017.02.086.

Point 2: Further research is needed because this method is not likely to solve their objective. At least, they need to clarify the weaknesses of this method and how they can achieve their goal in the future. It is necessary to clarify how to make the method clinically usable and what is lacking technically.

Response 2: We believe that after the major changes to the Title, Introduction, all Figures and Discussion, it is now clear what our main focus (mathematical modeling) is and how we contribute with advancements to one substantial piece (optimization algorithm) needed within a very complex imaging framework. Please see especially the last two paragraphs of the Discussion, where we now try to better explain who/what we follow up on, that it is still an experimental method, and what would need to be done to achieve application in clinical diagnostics.

Point 3: Figure 1 - This figure shows the overview of the research, so it is necessary to explain the content of the study. It is difficult to grasp the whole picture of the study design from this figure and legend.

Response 3: To make it easier for the reader to understand, we have split the original figure into two separate figures (Figure 1, Figure 3) and added more detail to the figure captions. Figure 1 now explains the representation of the cortical layers in T1 relaxation domain, while Figure 3 focuses only on the simulated data and how they differ in single-voxel and single-image experiments. The figures are referenced from different parts of the manuscript: Figure 1 is referenced from section 2.1 Fitting Problem and is also relevant when reading section 2.2 Experimental MRI Data; Figure 3 is referenced from section 2.3 Simulated Data and is also relevant when reading section 2.4 Modified Trust Region Algorithm.

Point 4: Figure 2 - Why is there a valley in the histogram at around 760ms? Since there is a peak of M0 at about 25ms, the signal in between is probably due to noise. Would you please clarify the meaning of this graph?

Response 4: This is now relevant to Figure 4. We have added an explanation of this phenomenon to the caption of Figure 4. A noticeable gap is present at 760 ms, indicating two separate distributions for the parameter estimates (for 700 ms and 800 ms, respectively). The distributions are skewed towards the estimation average.

Point 5: For equipment and software, you need to include the manufacturer's information.1)MRI scanner: e.g. (Siemens Healthcare, Germany) 2) MRiLab (Manufacturer, city, Nation).

Response 5: We have added the requested information: (i) 3T Siemens Magneton Prisma MRI scanner (Siemens Healthcare, Erlangen, Germany); (ii) MRiLab [22] simulator (Fang Liu, Madison, WI, USA).

Reviewer 3 Report

in file

Author Response

Dear reviewer, thank you for your valuable comments and insights which made us improve our manuscript significantly. Some of your comments correlated with those of Reviewer_2, and the most important ones led us to change the name of the manuscript to Identification of Laminar Composition in Cerebral Cortex Using Low-Resolution Magnetic Resonance Images and Trust Region Optimization Algorithm. We hope that we have responded to all your comments.

Point 1: The manuscript contains an interesting attempt to obtain quantitative information on various MR signal sources weighted with the T1 relaxation time. Additionally, an attempt to use this information to analyze patients' mental disorders is shown. Whether this approach is correct is another question that I will try to consider. Certainly the title is incorrect. There is no improvement in spatial resolution here. This is a misunderstanding. This is an approach similar to NMR relaxometric studies, e.g. rock cores (Overcoming barriers to .. shales porosity and similar issues) where using ILT (Inverse Laplace Transform) to analyze the NMR signal consisting of several or tens of thousands of echoes, we obtain data on the distribution of T2 times and then PSD. So the "indirect" analysis provides information about the nanostructures, while the signal comes from the entire sample. We are not talking about an increase in resolution here. Why not use ILT? (E. g. Enhanced Resolution Analysis for Water Molecules …)

Response 1: Thank you for the interesting analogies from other applications of MRI/NMR. We have changed the title to better reflect the main focus of our work. We agree that the method represents an indirect analysis. This indirect analysis allows us to obtain proportional representation of the cortical layers in the domain of spin-lattice relaxation, not the spatial domain. We are following up on the proven concept of low-resolution imaging of cortical layers presented in (Liftshits et al., 2018). Our goal is replicable imaging with sufficient sensitivity to identify the laminar composition of cortical layers, not a quantitative measurement. We believe that after the major changes in the Title, Introduction, all Figures and Discussion, it is now clear what our focus (mathematical modeling) is and how we contribute with advancements to one substantial piece (optimization algorithm) needed within a very complex imaging framework.

Point 2: Why you assume the existence of just 7 beans: T1 (700, 800, 1100, 1200, 1500, 1700, and 2000 ms) is not well explained anywhere.

Response 2: This is true – in the first version of the manuscript, we referenced the reader only to the literature (see e.g. [2] and [17] referenced in the Introduction). After the major revision, the seven layers in our model are explained in section 2.2 Experimental MRI Data (p5r173–178) . While the cerebral cortex is commonly delineated into six layers, our model assumes seven individual components. Given the resolution of our data, it is reasonable to assume the presence of additional bordering structures [cerebrospinal fluid (CSF) and white matter (WM)] within the cortical voxels. Therefore, up to seven unique components per voxel were assumed, each representing a unique formation within the cerebral cortex [CSF, WM, the sixth cortical layer, and the five remaining cortical layers].
[2] Economo, C. von; Triarhou, L.C. Cellular Structure of the Human Cerebral Cortex. 2009, I–XX, doi:10.1159/000226273.

[17] Barazany, D.; Assaf, Y. Visualization of Cortical Lamination Patterns with Magnetic Resonance Imaging. Cerebral Cortex 2012, 22, 2016–2023, doi:10.1093/cercor/bhr277.

Point 3: Experimental data. Considering TR and TE, we have different weightings of T1 and T2, i.e. some structures will indicate different T1 depending on the experiment. If these data are to be analyzed, they have to be normalized beforehand or it has to be shown (calculated) that such parameters (TR, TE) do not influence the measurement of the given structures.

Response 3: We are fully aware that sequence parameters can influence fitting results. However, keeping the parameters of the sequence consistent for all TI (neither shimming nor RF calibrations) should ensure reliable T1 estimation and stable and reproducible results of M0. We use an in-house modified pulse sequence to keep control over the sequence parameters. For instance, different T2-weighting and relatively long TE may lead to inaccurate estimation of the M0 parameter for each layer. Anyway, if the T2 of the component is known, this effect may be compensated. We consider spatial amplitude normalization for future real MRI data experiments and statistical evaluation. There are many other effects that influence the experiment. At this stage, we are not aiming at quantitative measurement yet. In an effort to replicate the low-resolution imaging experiments in (Liftshits et al., 2018), we have encountered a number of problems with the experimentally measured data and wanted to verify the ability to find the optimal model solution on data free from noise and other artifacts. Driven by the effort to reduce an unbearably large number of degrees of freedom, here we experiment mostly with simulated data and investigate whether this concept is viable and reliable. Having achieved promising results, we now believe that it is possible to achieve reliable relative differences between patients and healthy controls, which would be an excellent outcome in the future.

Changes in the manuscript related to this point and response: Introduction: p2r91-96, Discussion p11r361-370.

Point 4: The procedure for creating the reference data and using it to analyze experimental data is described in a highly unclear manner. Please describe in the table at the beginning of the results or the end of the methods: what are the standard-calibration data, what are they made of, how is the method of analyzing the experimental data carried out. What ROIs, what patterns, and what result. Various data is contained in the text, but it is difficult to trace its correctness. How practical is figure 1, what we see in it. Is this the whole brain T1 spectrum? If so, it would be good to show them e.g. for 2 volunteers and check if it looks similar. Additionally for 2 people with brain lesions and see if the spectrum is significantly different. What is the meaning of Figure 1?

Response 4: To make it easier for the reader to understand, we have divided the original Figure 1 into two separate figures (Figure 1, Figure 3) and added more detail to the figure captions. Figure 1 now explains the representation of cortical layers in the T1 relaxation domain, while Figure 3 focuses only on the simulated data and how they differ in our single-voxel and single-image experiments. Furthermore, we have performed some additional calculations to describe and demonstrate the relationship between parameter values in simulations and in the measured data, see new Figure 2 and the amendments in sections 2.2 Experimental MRI Data and 2.3 Simulated Data.

Regarding the sub-question related to previous Figure 1 – ”Is this the whole brain T1 spectrum?” The answer is no. If we had the entire brain T1 spectrum, we would have the job done as soon as the image data were acquired. Many more steps are needed, which should hopefully now be clear from the revised manuscript.

Point 5: Several fragmentary threads are described in the work. It should be emphasized that the formulation of the main idea and the main achievement in relation to the "state of the art" is more precise.

Response 5: We agree with this recommendation as we are really trying to improve the accuracy of the mathematical modeling that is an integral part of the whole method. Having revised the Introduction (p2r91-98) and Discussion (p11r357-377), our main idea and contribution should now be clear.

Point 6: Finally, I left a description of how to generate artificial, model data (equations 1-4). Where are we with our signal, in time domain, k space, or after the Fourier transform. The signal is complex in the time domain, after FFT we only have information about amplitudes (we see a map of proton-water densities, weighted relaxation times T1, T2, .., diffusion D). This description needs to be corrected.

Response 6: We have described this better in section 2.1 Fitting Problem: The Imaging protocol proposed in [18] produces only magnitude images after the inverse discreet Fourier transform. Hence, we have to limit the model to the magnitude data.

Point 7: The manuscript is potentially publishable, but requires many changes and clarifications. The idea is interesting and perhaps useful. This, however, in my opinion has not been communicated effectively, at least for the time being.

Response 7: First of all, thank you for your final encouragement. Having read your questions and the comments, we feel embarrassed to admit that we have not explained many aspects of our work adequately. We believe that with all the changes made to the text and figures, we have finally managed to clarify the whole manuscript.

Round 2

Reviewer 2 Report

The title has been revised to make it clear to understand the goal of this study. It would still need to have more discussion on making the approach feasible in the clinical field.

Author Response

Point 1

The title has been revised to make it clear to understand the goal of this study. It would still need to have more discussion on making the approach feasible in the clinical field.

Response 1

Thank you for your encouraging feedback on our first revision. We have added the following sentences into the Discussion section.

Further steps prior to clinical studies of a diagnostic application may include (i) evaluation of correlations between the identified MR imaging parameters of the cortical layers and selected variables resulting from histology of animal models, (ii) comparison with other established methods such as voxel-based morphometry, deformation-based morphometry, or diffusion kurtosis imaging. Given the possible clinical application in diagnostics, it is difficult to imagine this modality being used in isolation. Rather, it will be part of a multimodal approach to cortical pathology imaging, as it has the potential to provide complementary information on the internal cortex arrangement in pathologies that have a complex morphological correlate involving changes in elemental complexity, such as neuronal atrophy, dendritic tree reductions with increased density of neuronal bodies, migration of activated microglia, etc.

Reviewer 3 Report

1. The description of the equations is imprecise. Before the FFT we have a signal recorded by the receiving RF coil subjected to quadrature detection (Re, Im), then after the FFT we have an image composed of voxels of different intensity (corresponding to the proton density), weighted, among others, by the relaxation time - T1.
2. It is worth giving an analogy to ILT (some exemplary references showing, for example, the issue of PSD) in order to show the essence of the method - indirect obtaining information about the influence of various structures (components) on the observed signal.

Better written, publishable manuscript. I suggest adding the above explanations.
Good luck in your further research work.

Author Response

Dear reviewer, thank you for your encouraging feedback on our first revision. We believe that we have responded to all your minor comments.

Point 1
The description of the equations is imprecise. Before the FFT we have a signal recorded by the receiving RF coil subjected to quadrature detection (Re, Im), then after the FFT we have an image composed of voxels of different intensity (corresponding to the proton density), weighted, among others, by the relaxation time - T1.

Response 3.1

The descriptions of the equations have been updated to better specify their meaning. We have clarified the precise meaning of the “signal equation” as: “...  is complex-valued and represents single-voxel image information ...”. This is assuming monoexponential signal decay. The definition of the parameter has been specified as: “... the parameter denoting magnetization corresponding to the center of the k-space for the given voxel at  ...”. This definition should imply the correct specification of M0 – it reflects the whole initial magnetization and also the effects of all other sequence parameters which we maintain constant.

Point 2

It is worth giving an analogy to ILT (some exemplary references showing, for example, the issue of PSD) in order to show the essence of the method - indirect obtaining information about the influence of various structures (components) on the observed signal.

Response 2

We have added a paragraph (p3r132-140) explaining the context of the fitting problem from the mathematical point of view – as a problem of exponential analysis. We then introduce the inverse Laplace transform as a generalized solution to the problem of exponential analysis. Using its application in NMR relaxometry, indirectly estimating the contents of the measured sample, we illustrate the similarity within our work.